# EEG-Based Tool for Prediction of University Students’ Cognitive Performance in the Classroom

**DOI:** 10.3390/brainsci11060698

**Published:** 2021-05-26

**Authors:** Mauricio A. Ramírez-Moreno, Mariana Díaz-Padilla, Karla D. Valenzuela-Gómez, Adriana Vargas-Martínez, Juan C. Tudón-Martínez, Rubén Morales-Menendez, Ricardo A. Ramírez-Mendoza, Blas L. Pérez-Henríquez, Jorge de J. Lozoya-Santos

**Affiliations:** 1School of Engineering and Science, Mechatronics Department, Tecnologico de Monterrey, Eugenio Garza Sada 2501 Sur, Monterrey 64849, Mexico; mauricio.ramirezm@tec.mx (M.A.R.-M.); a00819192@itesm.mx (M.D.-P.); a00819083@itesm.mx (K.D.V.-G.); adriana.vargas.mtz@tec.mx (A.V.-M.); rmm@tec.mx (R.M.-M.); ricardo.ramirez@tec.mx (R.A.R.-M.); 2School of Engineering and Technologies, Universidad de Monterrey, San Pedro Garza García 66238, Mexico; juan.tudon@udem.edu; 3Precourt Institute for Energy, Stanford University, Stanford, CA 94305, USA; blph@stanford.edu

**Keywords:** EEG, cognitive performance, education, neuroengineering, machine learning

## Abstract

This study presents a neuroengineering-based machine learning tool developed to predict students’ performance under different learning modalities. Neuroengineering tools are used to predict the learning performance obtained through two different modalities: text and video. Electroencephalographic signals were recorded in the two groups during learning tasks, and performance was evaluated with tests. The results show the video group obtained a better performance than the text group. A correlation analysis was implemented to find the most relevant features to predict students’ performance, and to design the machine learning tool. This analysis showed a negative correlation between students’ performance and the (theta/alpha) ratio, and delta power, which are indicative of mental fatigue and drowsiness, respectively. These results indicate that users in a non-fatigued and well-rested state performed better during learning tasks. The designed tool obtained 85% precision at predicting learning performance, as well as correctly identifying the video group as the most efficient modality.

## 1. Introduction

University-level education is in constant evolution, making use of technological advancements to continue to provide high-quality pedagogical instruction to students. An important aspect of modern education is the contribution of information technologies, as different technological resources can be implemented to provide education under a variety of teaching modalities [1].

New learning modalities using modern technologies, such as dynamic lectures using live feedback from the students through wireless devices and social media educational platforms for student–teacher interaction, have shown increased acceptance rates in students when compared to more traditional teaching modalities [2]. Due to the increasing pace of technological innovation in education, teachers and students alike need to evaluate, adapt and adopt these new technologies and associated teaching modalities at a similarly increasing pace [3].

An extreme example of this need for adaptation of different teaching styles is the recent COVID-19 pandemic, where a vast number of professors and students around the world were forced to teach and learn under the online learning modality, many for the very first time [4]. This sudden change in education forced professors to change their conventional teaching approaches to face this challenge, with a steep adaptation learning process [4]. Different teaching modalities may have a different impact on students’ learning outcomes due to differences in perceived learning. This difference in perceived learning could be associated with personal learning preferences, as each student considers a particular learning style as the most effective [5]. Due to the need for fast adaptation to new learning modalities and differences in preferred learning styles in students, it is imperative that educators and researchers work together to develop tools and methodologies that provide a quantitative education evaluation under different learning approaches. The use of these tools could provide students with a variety of efficiently evaluated and validated learning approaches.

A promising technique to implement quantitative evaluation for learning modalities is electroencephalography (EEG), which is a non-invasive electrophysiological monitoring method that records the electrical activity from the brain at high temporal resolution across the scalp. The measured signals can be analyzed in the time domain, as in the case of event-related potentials (ERP) [6], as well as in the frequency domain, as the spectral content of frequency bands [7]. More recent EEG analysis methods involve the use of functional connectivity [8] and source separation [9] algorithms. Signal source separation methods for EEG signal analysis, such as the Moore–Penrose pseudoinversion proposed in [9], allow solving the inverse problem of neural recordings and spatially identifying different sources in the brain, responsible for specific neural activations.

In time domain analysis, the P300 wave, which arises under the presentation of an unexpected stimulus, has been used to determine the depth of cognitive processing in the brain [6]. Regarding spectral analysis, five EEG frequency bands are the most studied and well known. Each band has been associated with specific neurophysiological correlates. The delta (δ: 1–4 Hz) band is usually presented during periods of deep sleep, unconsciousness, anesthesia and lack of oxygen. Some reports have also associated this frequency band with different levels of cognitive load during mental tasks [7]. The theta (θ: 4–7 Hz) band primarily occurs in the parietal and temporal regions of the brain. Such waves are produced during moments of emotional pressure, interruptions of consciousness or deep physical relaxation [10]. The alpha (α: 8–12 Hz) band is usually observed at parietal and occipital regions of the brain when in conscious, quiet or rest states and its power decreases during active thinking. Reports in the literature have also associated changes in this frequency band with the generation of creative processes in the brain [11]. Beta band (β: 13–30 Hz) activity occurs predominantly in the frontal region during active thinking, sensory stimulation and alertness states [12]. Gamma band (γ: 30–50 Hz) activity has been linked to cognition and perceptual activity [13].

Analysis of EEG signals provides encouraging tools to evaluate and predict personalized cognitive traits in students, which can be used to gain a quantitative insight on learning outcomes. By analyzing high- and low-frequency EEG bands, information on the ongoing cognitive processes on students during learning can be obtained. High-frequency bands are useful at identifying alertness, active thinking, attention and multi-sensory processing states [13], while low-frequency bands can reflect relaxation, drowsiness and mental fatigue states [7]. The study of neural activity in the educational context is very recent, and it has sometimes been referred to as educational neurotechnologies [14]. This field of research is promising in understanding the neural traces of learning, as well as improving and enhancing learning for educational purposes. Due to this, educators and researchers are increasingly seeking for real-time measurements of neural activity in classroom environments [14].

Across the literature, there have been reports of analysis of EEG signals to predict intelligence and giftedness in children [15]. A study conducted by [15] focused on the use of EEG spectral features to detect gifted children in mathematics. In this study, students were defined as gifted if their scores on tests were higher than the group’s average. Epochs of the recording were labeled by the EEG equipment used in three mental states: workload, attention and relaxation. Mean, median, standard deviation, minimum and maximum values of these mental states were used as features for machine learning models, while the output was the classification of giftedness based on the mentioned criteria. The best machine learning model showed 76 % accuracy at correctly classifying giftedness.

In [16], a simultaneous EEG recording of twelve students was measured in the classroom over an eleven-class semester. EEG synchronization metrics were computed between the students and the teacher as well as enjoyment metrics. The results of the study showed that EEG metrics were correlated with students’ performance and enjoyability of the class. The results of the aforementioned studies suggest that cognitive performance classification and prediction can be obtained from analysis of EEG measurements.

As far as the authors know, few studies have presented pilot studies or methodological proposals to assess educational results at university-level education using EEG technologies to monitor cognitive performance on students in the classroom [17,18]. In [17], EEG signals of university students were measured during the solving process part of a ten-session computational thinking (CT) course. Students were divided into two groups, based on their curriculum and experience in CT: CT for the experienced and NCT for the inexperienced. The cognitive load of the students was estimated from the EEG signals, and the results showed, on average, lower cognitive load for the CT than the NCT group. In such study, the proposed analysis allowed identifying the increase in cognitive load as a higher challenge for the inexperienced NCT group than the CT group.

A recent study demonstrated the use of a neurofeedback tool called the attention monitoring and alarm method (AMAM), which continuously monitored the attention of students during e-learning sessions by analyzing their neural activity [18]. In such study, students using EEG headsets were provided feedback on their attention levels to help them regain focus in their learning sessions. The results of the study showed increased sustained attention and learning performance on the group using the AMAM, when compared to a control group not using it.

The use of highly portable (few channels, wireless) EEG devices has allowed researchers to explore different scenarios under the neuroengineering approach, in real-world settings. Examples of such equipment are the four-channel Muse [19], the three-channel Thinkmindset [20] and a reported low-cost, one-channel, Arduino-based brain–computer interface for brain state visualization [21]. A review on the use of portable EEG technologies in the educational context was presented in [22]. This review revealed that the use of EEG technology in education is a rather recent field; it is not more than one decade since the first related reports appeared. The review also stated that few studies (approximately 20) had been reported in this area of research. Therefore, the use of neurotechnologies for education is still an open research field which needs further development. This review notes that most of the education studies in neurotechnology make use of EEG equipment labeling algorithms such as the NeuroSky and the Emotiv systems which identify epochs of the recordings as cognitive load, high/low engagement, attention and meditation periods.

Although these metrics are useful when assessing learning outcomes, not all EEG equipment is provided with such labeling algorithms. Furthermore, the identification of emotional and other mental states such as frustration, mental fatigue, stress and anxiety is also useful to evaluate in the design of new teaching or learning approaches [23]. While not all commercial EEG instruments provide a reliable mental state classification, reports found in the literature show how these metrics can be calculated. Workload can be inferred from the mental fatigue measurement, where a high cognitive workload results in a higher mental fatigue state [24]. Mental fatigue can be described as a state where performance and attention decrease during cognitive tasks [10]. EEG processing methods oriented to mental fatigue detection were described in [10]. It can be observed in EEG by analyzing the theta/alpha ratio as well as the P300 wave [6,10].

A recent study on attentional markers of EEG in educational settings showed an increase in theta and decrease in beta bands for a prolonged attention period in the classroom [25]. These two markers can be combined into the beta/theta ratio to monitor real-time changes in cognitive processing capacity [26].

Neurofeedback tools for enhanced learning have also been reported across the literature. In [27], a neurofeedback protocol was implemented in musical training for children in order to increase the (θ/α) ratio at will. The results of the study showed increased musical creativity and overall skills in groups that received neurofeedback stimulation, compared to groups without feedback. A preliminary EEG feedback protocol was proposed in [28] to improve learning rates in distance education. The authors described the proposed protocols to monitor users’ mood based on α power calculation; however, no further analysis or results were reported.

An EEG tool that provides feedback to students about their attention level during a self-study task was implemented in [29]. An attention index provided by the used EEG equipment was provided as feedback to the students, and auditory feedback was given whenever the attention index decreased to a specific threshold. The experiments were implemented in an experimental group which received feedback, and a control group which did not receive feedback. The results showed a longer duration of attentive periods in the experimental group.

Based on all the aforementioned studies, it is clear that the use of neurotechnologies in education is a valid candidate to monitor and identify cognitive components during learning processes. However, most of the reported work in the field focuses on attention monitoring and feedback, and there is a gap in terms of predicting the performance of students, as well as using the neural activity of students to evaluate the characteristics of different teaching modalities. In this study, we propose the use of EEG measurements as a biomarker of cognitive traits to provide evaluation of different teaching and learning modalities, using machine learning models. The proposed model can be used to predict users’ performance on learning tasks, and to identify personalized optimal learning conditions. This prediction tool will also provide diverse applications in the educational field such as evaluation of different learning modalities. A neurofeedback protocol could also be implemented based on the results of the prediction model in order to give users a cognitive enhancement during learning experiences.

This paper is organized as follows: The methods implemented for the development of the machine learning tool are presented in Section 2. This section includes the experimental protocols, signal acquisition, pre-processing and analysis, feature extraction, model implementation and evaluation. A detailed description of the results obtained from this study and their discussion are presented in Section 3 and Section 4, respectively. Finally, the conclusions are presented in Section 5.

## 2. Materials and Methods

Volunteers were eligible for this study if they met the following selection criteria: participants had correct or corrected-to-normal vision (glasses), were enrolled in a university-level engineering course (the fifth to the eight semester) and were under no medication at the time of the experiment. Experiments took place in the library of the Tecnologico de Monterrey in a semi-closed cubicle area, at a time between 10:00 a.m. and 16:00 p.m., depending on the personal schedule of the volunteers. The methods and a preliminary study on the data collected from this study were presented in [30].

A total of 20 healthy volunteers participated in this study: 10 (5 male, 5 female) in the text group and 10 (7 male, 3 female) in the video group. Average ages were: ($\mu_{T}=22.3\pm 1.63$) for the text group and ($\mu_{V}=22.7\pm 2.26$) for the video group. Data from one participant in the text group were discarded due to problems in the measurements. Before starting the experiments, a consent form was handed to each participant with a detailed description about the experimental procedures of the tasks to perform. Volunteers were asked to sign the consent form if they were willing to continue with the experiments. Volunteers also agreed to be recorded in pictures or video by signing this form. EEG signals were acquired for all subjects during three different stages of each experimental trial: 30 s of eyes closed (EC), 30 s of eyes open (EO) tasks and during performance of the learning tasks. Learning tasks consisted of a 150-s presentation of either the text or the video material depending on the experimental group each participant belonged to. Both groups performed three consecutive learning trials, with resting periods of two minutes between recordings. The content itself of the learning trials was the same across groups, with the exception that the text group learning task consisted in reading a plain text, while the video group learning task consisted in an audiovisual presentation of the same information. Volunteers in the text and video groups were asked to read and watch silently to avoid unwanted artifacts in the EEG signals due to movement from the tongue and facial muscles. However, to ensure all volunteers acquired the knowledge presented in the learning trials, they were asked to answer standardized tests with questions about the given topic after each recording. The topic presented during learning trials is that of controller area network (CAN) bus, a type of communication protocol employed within vehicles for efficient vehicle-related data transfer. This specific topic is part of the subject “Automotive Electronics” of the mechatronics engineering program at Tecnologico de Monterrey.

The information presented to the students in a specific group was the same during the three learning trials (LT), i.e., three repetitions of the same text or the same video. However, the evaluations for each LT were different for each repetition. The evaluation consisted of tests with an increasing number of questions related to the presented information during learning trials. These evaluations consisted in three, five and nine questions, respectively, for the first, second and third repetitions. The difficulty of the questions also increased from the first to the third repetition.

A visual representation of the proposed experimental setup is presented in Figure 1. Information presentation and evaluation protocols for all learning tasks were designed in Spanish, as it was the native language of all participants. Original texts, videos and tests and those translated to English used during learning tasks and evaluation are presented in the Appendix A.

### 2.1. Signal Acquisition and Analysis

For this study, EEG signals were acquired wirelessly using the OpenBCI system. The Ultracortex Mark IV headset was used for EEG acquisition, which makes use of highly portable and mobile dry electrodes. This headset includes a combination of eight EEG sensors distributed across the scalp, as well as accelerometers measuring three-dimensional acceleration. The electrodes’ placement follows the international 10–20 reference system, including two frontal, two central, two parietal and two occipital electrodes, respectively: FP2, FP1, C4, C3, P8, P7, O2 and O1. The OpenBCI headset used is shown in Figure 1.

These signals were recorded and wirelessly transmitted to a computer through the data acquisition, processing and design tool OpenVibe. All data were acquired at a sampling frequency of 256 Hz and filtered with a 0.1–100 Hz, 4th-order Butterworth bandpass filter. All EEG signals were cleaned using the artifact subspace reconstruction (ASR) algorithm using a parameter, κ=15, to reduce large artifacts. ASR is an effective and efficient signal cleaning method that reconstructs artifacts as large as κ times the standard deviation of a clean portion of the signal. This value of κ was selected as values between 10 and 100 are suggested to effectively remove muscle and eye movement-related activity, while not being as aggressive in removing important EEG-related activity [31]. EEG signals were used to calculate power in five frequency bands: delta (1–4 Hz), theta (4–7 Hz), alpha (8–12 Hz), beta (13–29 Hz) and gamma (30–50 Hz). Power was calculated in one-second windows using the fast Fourier transform for all EC tasks, EO tasks and LTs in all channels and frequency bands. Power values were normalized to EO tasks for all users, following Equation (Equation 1)
(1)$$P_{N}{\left(t\right)}_{ch,fb}=\frac{P_{LT}{\left(t\right)}_{ch,fb}-{\bar{EO}}_{ch,fb}}{{\bar{EO}}_{ch,fb}},$$
where ${\bar{EO}}_{ch,fb}$ represents the average power at specific channels (ch) and frequency bands (fb) during EO tasks, and $P_{LT}{\left(t\right)}_{ch,fb}$ represents the power values across time at the same channel and frequency band during an LT before normalization. This procedure allowed obtaining a second-to-second normalized power estimate of the EEG signals in the five mentioned frequency bands. By averaging the signals obtained by implementing Equation (Equation 1), average normalized power values for all channels and frequency bands were used as features in a linear regression model (5 frequency bands × 8 channels =40 features).

Power ratios were also calculated for all possible combinations of the analyzed frequency bands. Equation (Equation 2) shows the calculation for the power ratios.
(2)$$P_{R}{\left(t\right)}_{ch,(A/B)}=\frac{P_{LT}{\left(t\right)}_{ch,A}}{P_{LT}{\left(t\right)}_{ch,B}},$$
where $P_{LT}{\left(t\right)}_{ch,(A)}$ and $P_{LT}{\left(t\right)}_{ch,(B)}$ represent the power across time (before normalization) during an LT at channel (ch) and frequency bands *A* and *B*, respectively. Then, average power ratios were used as features for the linear regression model (20 ratios × 8 channels =160 features). All the ratios, obtained using Equation (Equation 2), were calculated for the three repetitions. All frequency bands and power ratios considered in this study are represented in Table 1.

A total of 200 features were obtained as variables for the multivariate linear regression model (MLR): 40 (normalized EEG power) + 160 (EEG power ratios). Figure 2 shows a representation of the obtained EEG signals after performing the pre-processing methods. Representative EEG signals of participants $Pt_{1}$ and $Pv_{4}$ from the text and video groups, respectively, are presented, during the first five seconds of the EC task, EO task and LT of the first repetition. Average normalized α/θ ratios across the scalp, corresponding to each five-second EEG recording, are also presented in Figure 2, as well as the topographical distribution of the available electrodes in the used EEG system.

### 2.2. Statistical Analysis

In order to observe differences between the two evaluated learning modalities, a statistical analysis was implemented as follows. Two-sample *t*-tests (p<0.05) were performed to observe statistically significant EEG power differences between text and video groups across different learning stages. Statistical tests were performed at all repetitions. All channels, frequency bands and power ratios were considered for this analysis. A total of 600 statistical tests were performed between all subjects of both groups: (8 channels × 5 frequency bands × 3 repetitions) + (8 channels × 20 ratios × 3 repetitions). In this analysis, *p*-values indicate the probability that observed changes in the means of both groups are chance-related. Therefore, significance suggests that the observed changes are related to the experimental differences of each group (learning modality).

A pair-wise Pearson correlation analysis was implemented in order to find the most relevant features that show a correlation with users’ scores on the test. All scores from the three repetitions of both text and video groups were included in each correlation test and treated as unlabeled data. Each test contained data from all subjects and all repetitions on both text and video groups. A total of 200 correlation coefficients and *p*-values were calculated between pairs of 60 scores and 60 values of the analyzed feature, testing the hypothesis of no correlation between variables. In these tests, *p*-values represent the probability that the observed correlation is obtained by chance. Therefore, small *p*-values suggest that the correlations between the evaluated variables are significant. This analysis was used to implement a dimension reduction on the MLR model; thus, only the most correlated features (correlations which presented *p*-values <0.05) were selected to be used in the MLR model.

### 2.3. Model Evaluation

From the correlation tests, the significant features were included in the MLR models. A feature matrix was built using the average (during LT) of all the significant features for all participants and repetitions. Features in this matrix were sorted according to their *p*-value obtained in the correlation tests. All MLR models were evaluated using a 70:30 data ratio for training and testing sets, respectively.

The MLR model was trained using 42 samples from the selected features. Each model was trained and evaluated in an iterative process with an increasing number of features, from 1 to *K*, where *K* is the maximum number of features for that specific model. Then, each model was used to predict the students’ scores in the test, using the remaining 18 samples for a 10-fold cross-validation. At every iteration, the parameters of the MLR models were calculated using the normal equation method. At every iteration, different random data were selected for both training and testing sets, and models were built and evaluated. Following these criteria, data from participants in the training set were not included in the test set. The predicted values were compared to the real scores, and accuracy was calculated as presented in Equation (Equation 3):(3)$$Acc_{k}=100-\left(\right|{\bar{PS}}_{k}-{\bar{RS}}_{k}\left|\right),$$
where $Acc_{k}$ is the average accuracy of an MLR model using *k* features across 10 cross-validation tests, and $|{\bar{PS}}_{k}-{\bar{RS}}_{k}|$ represents the average absolute difference across cross-validation tests between the predicted (PS) and real scores (RS) at that specific model. Accuracy was evaluated for MLR models using different numbers of features to find the most reliable model. Signal analysis, feature extraction, selection and model evaluation were implemented by custom codes in Matlab 2020a version 9.8.0.1396136 (The MathWorks Inc, Natick, MA, USA).

A flowchart representing the complete methodological process implemented in this study is presented in Figure 3.

## 3. Results

### 3.1. Learning Performance

Table 2 shows the average scores and exam times (ET) across subjects obtained for both groups. Scores and times are shown for all three repetitions. The percentage of improvement for each repetition is also presented in this table for scores and exam times. The score and time improvement metrics are defined as presented in Equations (Equation 4) and (Equation 5), respectively:(4)$$SI_{r}=SV_{r}-ST_{r},$$
(5)$$TI_{r}=\left(1-\frac{TV_{r}}{TT_{r}}\right)*100,$$
where $SI_{r}$ and $TI_{r}$ represent the score and time improvement, respectively. Here, a positive improvement means there was a higher performance in the video group than in the text group at repetition *r*, while a negative improvement stands for the opposite case.

Table 2 shows that the video group had a better performance and improvement after each repetition in both the ET and scores, except in the scores during the first repetition, where both groups obtained an equal performance. These performance differences are also observed in Figure 4. Both groups started with the same average score test in repetition one, 83.33 (out of 100), and showed a slight increase (15%) in time improvement, 34.5 s for the video group and 40.8 s for the text group. In repetition two, the video group had a better performance in both the score (80) and time (48 s), while the text group’s average score was 76, with a time of 62.3 s. The video group obtained, on average, a 5% higher score than the text group, and a 22% time improvement. In repetition three, there is a higher difference between the results of both groups, where the video group obtained an average score of 94.7 and a time of 78.5 s, while the text group had a score of 76.67 and a time of 103.6 s. In this last repetition, the video group obtained improvements of 19% and 24% in the scores and ET, respectively. These results indicate that the video group obtained a better performance overall when compared to the text group. Furthermore, the improvement consistently increased as repetitions went on. This means performance differences between groups were more notorious at more difficult trials.

### 3.2. Statistical Analysis

Figure 5 shows the channels which passed the significance tests, found at each frequency band, for all repetitions. Figure 6 shows the channels which passed the significance tests, found at each power ratio, for all repetitions. In these colormaps, a positive value indicates a higher power (or power ratio) in specific channels and frequency bands in the video group than the text group. Negative values indicate a lower power (or power ratio) in the video group than the text group at specific channels and frequency bands.

During repetition 1, the video group showed lower power in beta and gamma in parietal and occipital electrodes, as well as higher power in the theta band (O2). During repetition 2, the video group obtained, in general, higher power in frontal electrodes, in δ, θ, α and β bands, while the group obtained lower occipital power in the θ and β bands. During the third repetition, the video group showed, in general, lower power values at occipital electrodes in δ and θ bands and centro-parietal electrodes in β and γ. There was also higher frontal (FP2) power in the alpha band in the video group than in the text group.

In the colormaps presented in Figure 6, positive values in upper rows mean that the video group showed a higher proportion of high-frequency bands over lower-frequency bands when compared to the text group.

Positive values in lower rows mean that the video group showed a higher proportion of lower-frequency bands over high-frequency bands when compared to the text group, and vice versa. When colormaps show negative values, this indicates that the text group showed higher values in specific power ratios and channels than the video group.

During repetition 1, the video group showed a higher activity of lower-frequency bands over high-frequency bands, while the text group showed a higher activity in higher-frequency bands over lower-frequency bands. This can be observed in Figure 6 as positive values in lower rows, as well as negative values in upper rows of the colormaps. A similar behavior was observed in repetition 2 for a smaller amount of channels, and considerably less significant changes were found at the third repetition. In general, these changes were observed in parietal and occipital electrodes, as well as a few central and frontal electrodes, in repetitions 1 and 2, respectively. Significant differences were found mostly in the (θ/α), (γ/β), (γ/θ), (γ/δ) and (γ/α) power ratios.

Table 3 shows the number of statistically significant channels found using the described statistical analysis, per frequency band and power ratio. An increasing trend of significant changes was observed on frequency bands: 6 for δ, 14 for θ and α, 19 for β and 20 for γ. It can be observed that a higher amount of significant changes were found in features related to an increased activity of higher-frequency components. Differences between the evaluated learning modalities were reflected as few changes in lower-frequency components, and a higher amount of changes in higher-frequency components, mainly in occipital and parietal regions.

### 3.3. Correlation Analysis

The results of the implemented correlation analysis show only six features as significant. These features were considered as the most relevant and were selected as features for a dimensionally reduced MLR model. The selected features, as well as their *p*-values and correlation coefficients, are presented in Table 4. The selected features were: C3 (α/θ), C3 (θ/α), FP1 (δ), C3 (γ/α), P8 (α/γ) and P8 (γ/α). Figure 7 shows the negative correlation between the (θ/α) ratio observed at C3 and students’ scores, and Figure 8 shows the negative correlation between δ power found at electrode FP1 and students’ scores.

### 3.4. Model Evaluation

An MLR model was designed and evaluated to predict students’ scores based on the selected most relevant features presented in Table 4. The selected features were sorted according to the obtained *p*-value in the correlation tests. Therefore, the first feature was that which obtained the lowest *p*-value, etc. Features C3 (θ/α) and P8 (γ/α) were considered as redundant because their inverse were already selected as relevant and thus removed from the following procedure. The model was trained and evaluated using an increasing number of features k=1…K, where K=4 is the maximum number of features. Figure 9 shows the average accuracy across 10 cross-validation iterations of the MLR model at predicting students’ scores, for MLR models using k=1…K features. A consistent average accuracy of 85% was observed for MLR models using different numbers of features. The highest accuracy was found for the model using two features, at 85.76% accuracy.

This model was used to predict the scores of 30 different random subsamples of the same size as in the cross-validation tests. The predicted scores were labeled according to the real groups (text or video) they belonged to. Then, the labeled predicted values were compared to the real values in both groups. Figure 10 shows the average real and predicted scores for all the evaluated data samples in this analysis, in their respective groups.

A considerably small difference between the real and predicted scores was observed in both groups. In the real data, the average scores of text and video were 81.23% and 86.36%, respectively, while the average predicted scores were 81.15% and 86.32% for the text and video groups, respectively. In both cases, the video group obtained, on average, higher scores than the text group. With this analysis, it was observed that the designed machine learning tool, based on EEG-related features, correctly predicted higher student scores in the video group than in the text group during learning trials.

### 3.5. Factor Analysis: Group and Difficulty

A more detailed statistical analysis was performed on the data to explore the effects of different factors, such as the learning modality (group) and the repetitions (difficulty), on the learning performance and selected features.

For this analysis, the non-parametric Friedman variance test was applied, organizing the data by columns (group) and rows (difficulty) with 10 replicates (participants) in each block (p<0.05). The null hypothesis of this test is that average performance values (ET and score) and selected features between groups and difficulties do not change significantly. If the test detects significant changes to a specific factor, then the null hypothesis is rejected. Eight tests were performed: one for ET, one for score and one for each of the features presented in Table 4. The results of these statistical tests are presented in Table 5.

The results of this analysis show that the group factor had a significant effect on the scores, ETs and features C3 (α/θ), C3 (θ/α) and FP1 (δ). The difficulty factor also had a significant effect on the same features; however, it did not have a significant effect on scores and ETs. The remaining features, C3 (γ/α), P8 (α/γ) and P8 (γ/α), were not affected significantly by any of the analyzed factors. An interesting observation is that by changing the conditions of the learning task, the performance of the participants, as well as their neural activity, changed significantly. This implies that although the contents were the same for both modalities, the modality of their presentation takes an important role in defining the learning outcome. This difference in learning modality significantly changed not only the learning performance but also the brain activity of the participants.

Boxplots showing the average distributions of the scores, ETs and feature C3 (α/θ) in the text and video groups, as well as during different difficulty levels, can be observed in Figure 11.

In Figure 11, lower ETs and C3 (α/θ) and higher scores were observed, on average, for the video group compared to the text group. A notorious trend was observed for the difficulty factor. On average, a decrease in C3 (α/θ) values, as well as an increase in ETs and scores, was observed across difficulties. The increase in ETs and the decrease in C3 (α/θ) suggest a relation between the development of mental fatigue across the different repetitions of the learning tasks. An interesting increase in scores was observed after the third repetition, which can be explained by those in the video group, as shown in Figure 4.

## 4. Discussion

Regarding the performance analysis, the results indicate that multi-sensory (video) conditions resulted in higher performance benefits for the students when compared to the performance of the students performing unisensory (text) tasks. The key difference between unisensory and multi-sensory training exists during encoding, whereby a larger set of processing structures are activated in the multi-sensory paradigms [32]. Multi-sensory exposure enables stimuli to be encoded into multi-sensory representations and, thus, will later activate a larger network of brain areas, facilitating learning [32]. The evaluation shown in Table 2 suggests that audio-visual stimulation when studying contributed to a more successful learning experience than text reading.

The statistical analysis allowed obtaining further insight on the spectral differences between EEG signals during both learning modalities. In general, the video group showed a decreased higher-frequency EEG activity when compared to the text group. This is observed as negative values in Figure 5 and Figure 6 mainly in β and γ bands. This is also represented in Table 3. The decrease in beta and gamma activity can be result of the phenomena previously observed in the context of (1) motor learning and reduced preparatory or attentional demands, and (2) memory encoding, respectively [13]. Another factor that can be a cause of the decrease in high-frequency EEG activity of the subjects in video groups is that multi-sensory enhancement can take on several forms in EEG activity, including: increase in firing rate, resetting the phase of ongoing oscillatory activity and decreasing response latencies. Furthermore, each of these mechanisms could have the effect of enhancing plasticity [32]. Apart from the decreased activity level of high-frequency waves, the synaptic plasticity fostered by gamma oscillations would help in selecting the most relevant processes and lead to the creation of a sparse and faster neural route, resulting in a sharpened representation of the learned stimulus [13]. This is another factor that can be taken into account to suggest that students in the video group had a better learning experience, and this resulted in a better performance during the evaluation phase.

We suggest that the decreased levels of β and γ activity in the video group when compared to the text group are a reflection of an optimized use of resources of the neural circuitry. In the text group, students needed a more increased focus and attention level than students in the video group in order to successfully encode and retain the presented information, causing an increase in high-frequency brain activity. This difference in learning techniques was, in turn, reflected as increased β and γ activities for students in the text group. Moreover, most changes were found in the occipito-parietal region. This most likely represents the differences in visual stimulation in the occipital region between the two types of evaluated learning tasks [10].

In [13], the authors reported that γ power decreases as a skill is acquired by a subject. It was also mentioned in their conclusions that an apparent α power increment was observed during this same process. This might imply that in the initial learning stages where users need to focus more on the tasks at hand, there are high γ power and low α power—in other words, a low (α/γ) ratio. Otherwise, during the late stages of learning, where users do not need to focus as much to perform the tasks, there are low γ power and high α power, or a high (α/γ) ratio. This reasoning suggests that subjects showing high (α/γ) values are in a more efficient cognitive state than subjects showing lower (α/γ) values, which, in our case, reflects into higher performance scores during learning tasks.

The correlation analysis revealed the most relevant features when predicting performance on learning trials were C3 (α/θ), FP1 (δ), C3 (γ/α) and P8 (α/γ). It is interesting to mention that the (θ/α) ratio has been previously associated with the degree of mental fatigue in subjects [10]. As observed in Figure 7, higher values were observed when users performed poorly. This behavior is consistent with reports of how mental fatigue affects task, as well as cognitive, performance [10]. Furthermore, Figure 8 shows the negative correlation found between delta power (which is associated with sleepiness) and students’ scores. Based on these figures, it can be inferred that students in a mentally fatigued and drowsy state were more prone to obtain lower scores on the tests. On the other hand, users that obtained higher scores on the evaluation showed a low (θ/α) and delta power, which are indicative of a non-fatigued and well-rested state. Another interesting finding was the positive correlation between the (α/γ) ratio and students’ scores, observed in Table 4. Given the positive correlation, it is logical to think that higher (α/γ) values were observed for students with higher scores. Furthermore, in Figure 6, it is shown that in all repetitions, the (α/γ) ratio is significantly higher in the video group than in the text group which, as observed in Figure 4, obtained, on average, higher scores as well.

It seems that the used features in the MLR model allowed predicting students’ scores and correctly identifying which study technique would give better performance results, based exclusively on EEG-related features. The use of other physiological signals such as skin temperature, electrodermal activity and heart rate can further increase the features in the model in order to increase its accuracy. These types of features have been used to build models that help predict mental fatigue and drowsiness in subjects [33]. Another worthy addition to the system is the use of a P300-inducing protocol to observe amplitude and latency changes due to cognitive impairment [6,10], which could be helpful to evaluate burnout effects induced by specific teaching strategies. The presented results are promising, as the presented implementation could be used by students desiring to improve their self-study skills [29]. It is also a potential tool of interest for educators, in order to evaluate the cognitive performance of different learning and teaching techniques.

Interestingly, in Figure 11, lower C3 (α/θ) values are observed for the video group compared to the text group. By taking a closer look at the average C3 (α/θ) values across groups and difficulties, it can be observed that they are relatively similar, except for trial 2 of the video group, which shows the lowest average. Although this effect is not significant, as observed in Table 5, this could be interpreted as a slight increase in workload in the video group, probably due to the stimulation presented in the learning trials. However, even in this state, volunteers obtained a better learning performance. The use of this type of methodology is therefore useful to identify advantages and disadvantages in different types of learning modalities and could be used by educators to evaluate teaching approaches in a quantitative manner. The ideal design of a teaching approach would be that which does not induce a high-mental fatigue state in students while, at the same time, helping them obtain a higher learning performance.

### Limitations

One limitation of this study that needs to be addressed is the small sample size. Only ten volunteers were recruited for each group, and this could question the reliability of the results. Due to the sample size, it could be argued that the differences observed between groups could be strongly affected by the individual state of the users rather than the condition. However, by feeding the proposed model with information from both groups, the resultant model provided a more general representation and could predict the cognitive performance of students at different conditions efficiently. Nonetheless, and as mentioned previously, the sample size is one of the limitations in the study and will be addressed in future research.

Although the observed patterns in the data and the obtained results allow proposing a logical connection of thoughts about changes in mental states and cognitive performance during learning tasks, a larger sample size could provide more insight on the results. The authors’ interest is to extend the present study in future work to a bigger sample size to obtain more statistical confidence. The enlargement of the database will also allow the authors to develop and test more complex algorithms such as deep neural networks [34], which have been applied in systems that provide a reliable classification of mental states using EEG signals as input. Such algorithms need a considerable amount of data to effectively model the desired phenomena, which is yet another motivation for increasing the sample size.

## 5. Conclusions

The present study described protocols that demonstrate how neuroengineering tools, along with machine learning techniques, can be used to design models that predict users’ performance during learning tasks. The results presented in this study suggest that performance differences in learning techniques might be reflected by differences in EEG spectral components. In this study, reduced activity of high-frequency components during an interactive, video-watching learning modality was observed, probably as a reflection of an optimized use of cognitive resources. In a traditional text-reading learning scenario, there was an increased activity of high-frequency components, related to active thinking and information processing, suggesting an increased effort in these types of learning modality. However, in this latter case, the involved cognitive processes were not as effective in encoding and retaining information, resulting in lower performance scores than those obtained by the video group.

It was also observed that users showing features indicative of a non-fatigued and well-rested state obtained a higher performance than users showing mental fatigue or sleepiness brain patterns. The proposed model, which was implemented as an offline method, was effective at predicting students’ scores during learning trials based on four EEG-related features, with 85% accuracy.

The presented model is a simple, yet effective method at predicting students’ cognitive performance in the classroom, by monitoring and interpreting their EEG signals. Future research will be oriented to developing an online version of the designed machine learning tool, which can be used to provide real-time monitoring and neurofeedback under different learning paradigms. These tools might be helpful to students, teachers and educators in evaluating the cognitive responses to various traditional and more innovative learning and teaching modalities. Such a system can also compare the effectiveness of different teaching modalities (and mental states of students) before, during and after lockdown due to Covid-19, such as face-to-face lectures and teamwork activities [35], distance-learning (synchronous, live-streaming classes; asynchronous, pre-recorded classes) [36], massive online open courses (MOOCs) [37], flipped classrooms [38] and hybrid modalities [39], among others.

## Figures and Tables

**Figure 1 brainsci-11-00698-f001:**
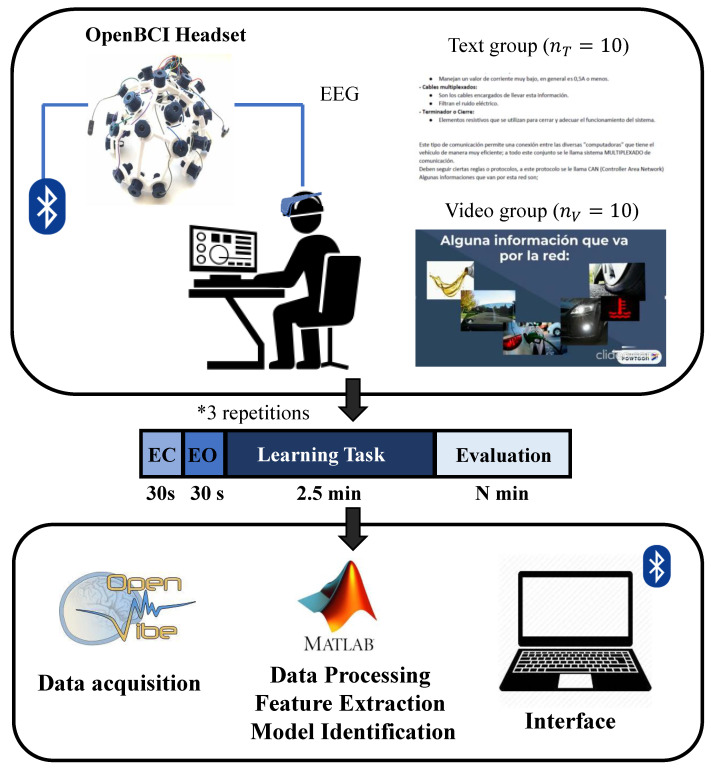
Proposed methodology for implementation of a cognitive performance predictive tool. EEG recordings were measured for text and video groups during baseline (EC, EO), learning tasks and evaluation for three repetitions. EEG data were transferred to a PC via OpenVibe and analyzed using Matlab to build the predictive models.

**Figure 2 brainsci-11-00698-f002:**
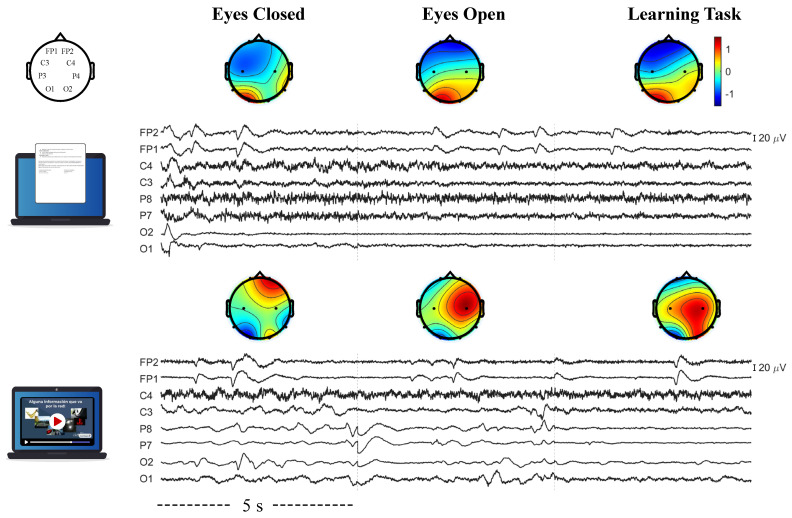
Representative EEG recordings during the first five seconds of eyes open, eyes closed and learning tasks for text (**top**) and video (**bottom**) groups, measured at electrodes: FP1, FP2, C3, C4, P7, P8, O1 and O2. The spatial distribution of EEG electrodes is shown in the top left inset. Topographic plots show the average α/θ ratios (normalized between −1.5 and 1.5), computed from each five-second EEG recording for all electrodes. The presented EEG signals correspond to those of participants $Pt_{1}$ and $Pv_{4}$ for the text and video groups, respectively, during the first LT repetition.

**Figure 3 brainsci-11-00698-f003:**
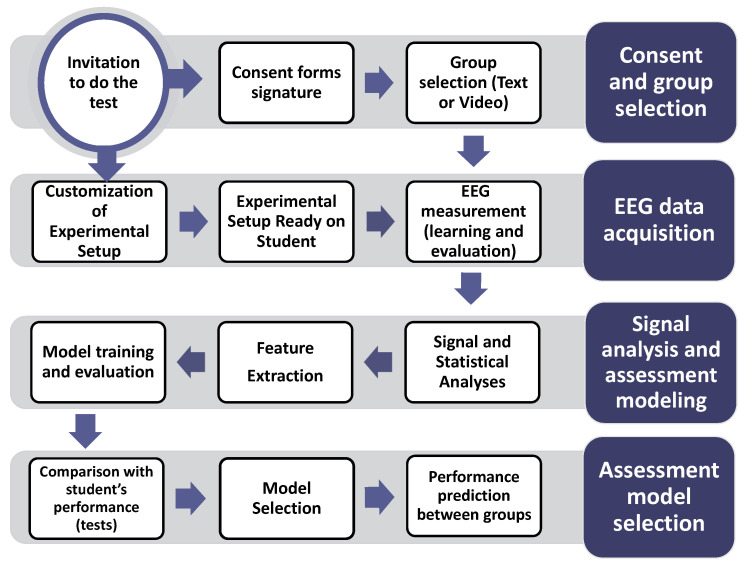
Flowchart of the experimental protocol implemented in this study. Participants were invited to the study and, after providing consent, performed the learning tasks. After EEG signals were measured and analyzed, relevant features were used to build and evaluate models.

**Figure 4 brainsci-11-00698-f004:**
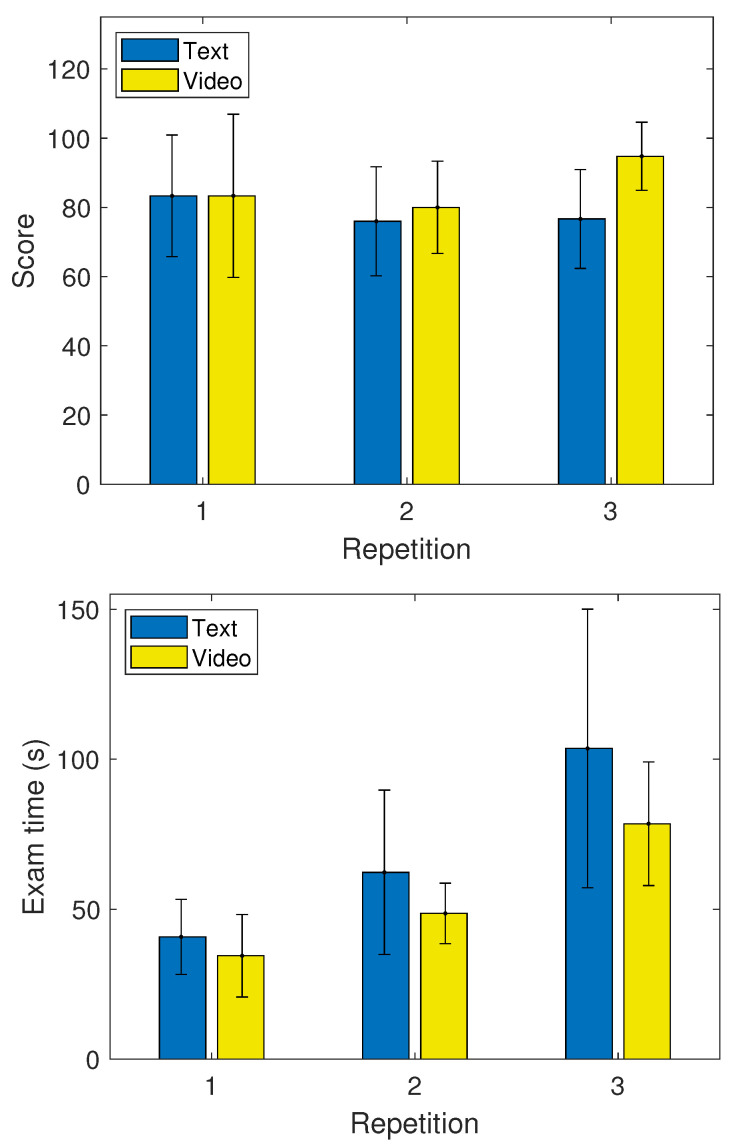
Average scores (**top**) and exam times (**bottom**) for text and video learning trials for all three repetitions. Black lines represent standard deviation.

**Figure 5 brainsci-11-00698-f005:**
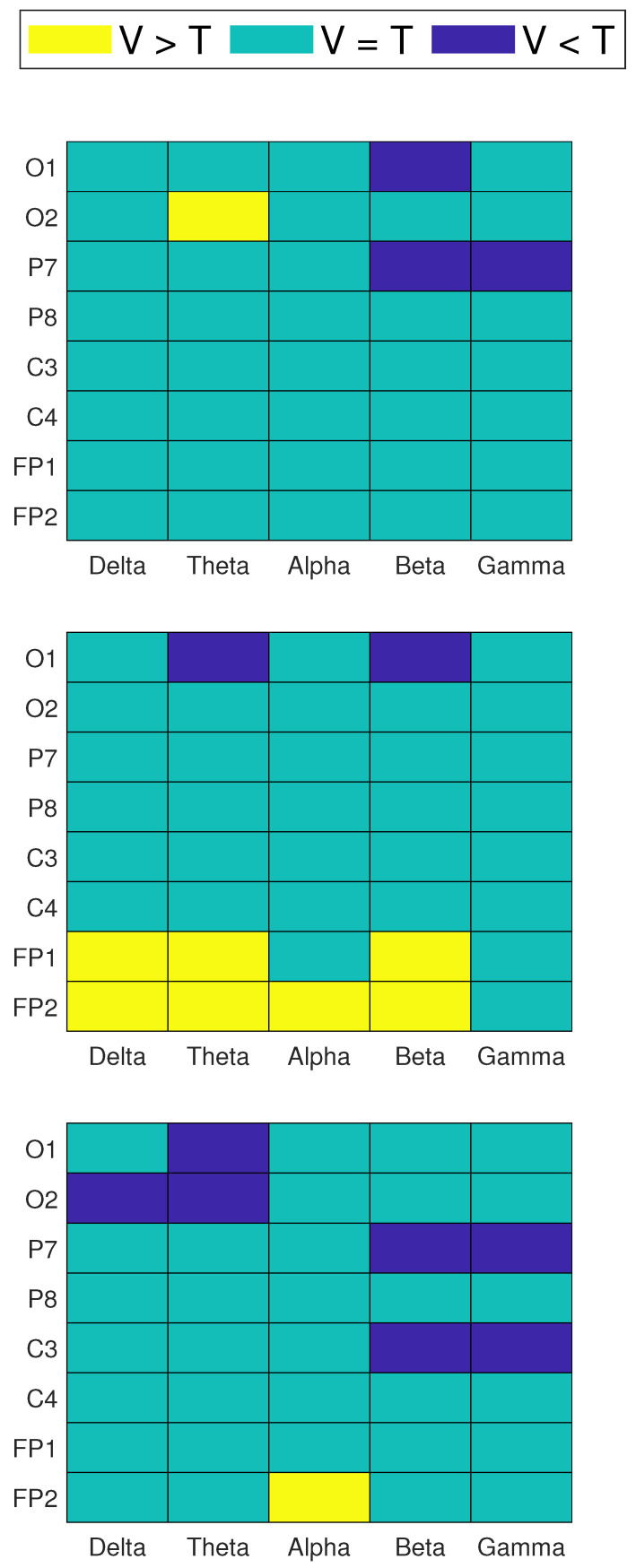
Channels and frequency bands which showed statistically significant (p<0.05) higher power during video compared to text (V > T), lower power during video compared to text (V < T) and no significant changes between groups (V = T). Three colormaps represent the three LT repetitions: 1 (**top**), 2 (**middle**) and 3 (**bottom**).

**Figure 6 brainsci-11-00698-f006:**
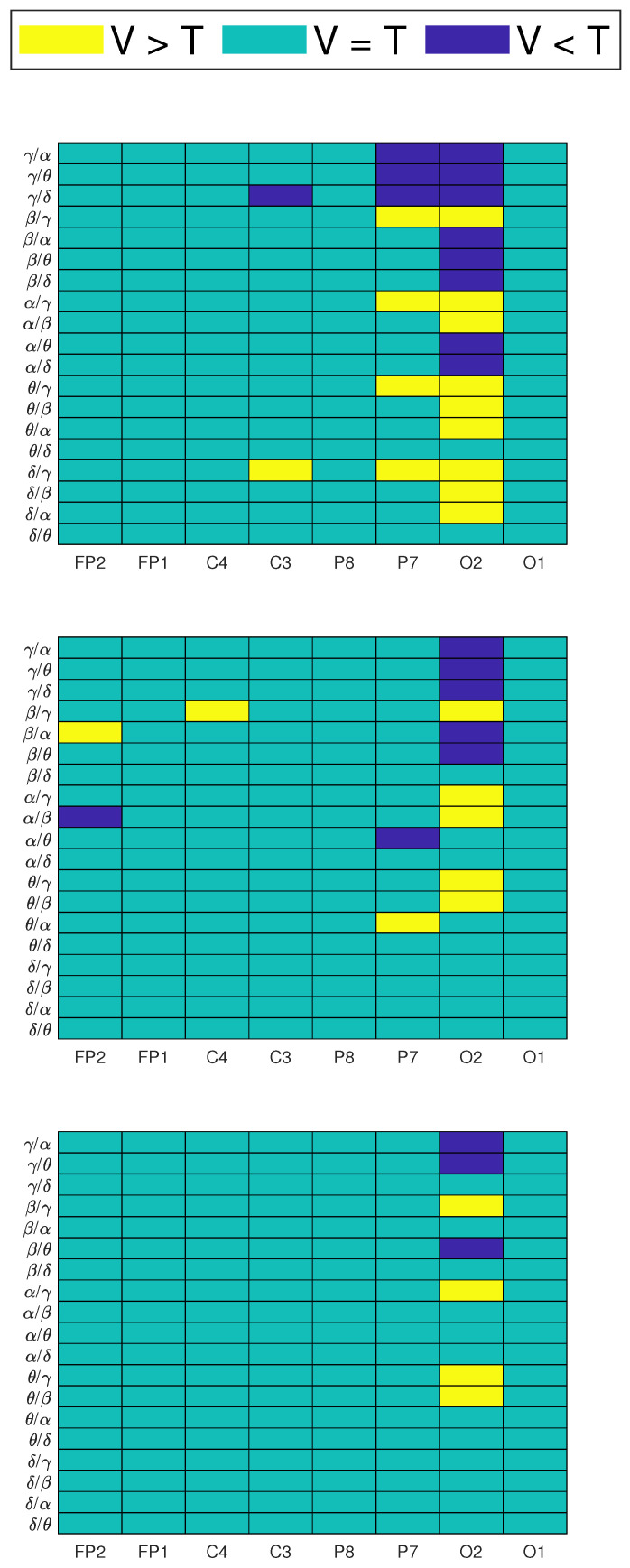
Channels and power ratios which showed a statistically significant (p<0.05) higher power difference during video compared to text (V > T), lower power difference during video compared to text (V < T) and no significant changes between groups (V = T). Three colormaps represent the three LT repetitions: 1 (**top**), 2 (**middle**) and 3 (**bottom**).

**Figure 7 brainsci-11-00698-f007:**
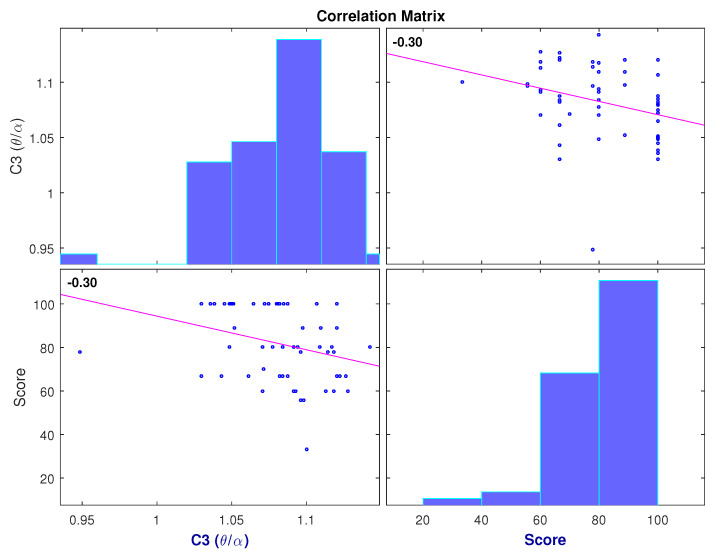
Correlation between C3 (θ/α) ratio and test scores for all participants from both video and text groups, at all LT repetitions.

**Figure 8 brainsci-11-00698-f008:**
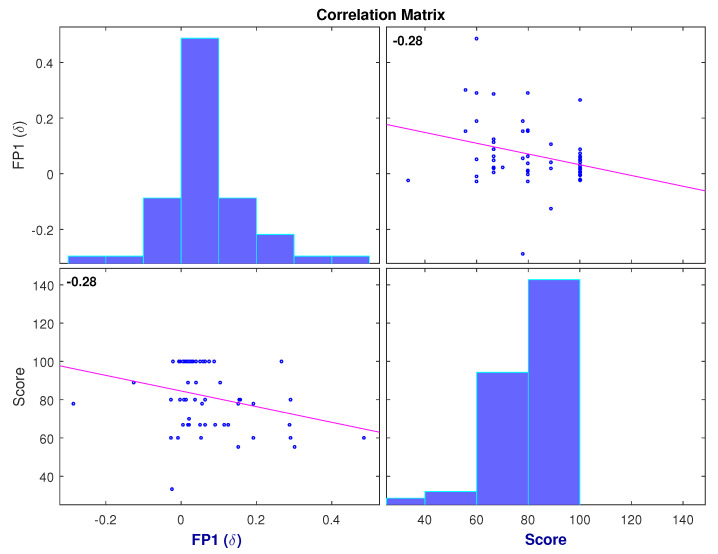
Correlation between normalized FP1 (δ) power and test scores for all participants from both video and text groups, at all LT repetitions.

**Figure 9 brainsci-11-00698-f009:**
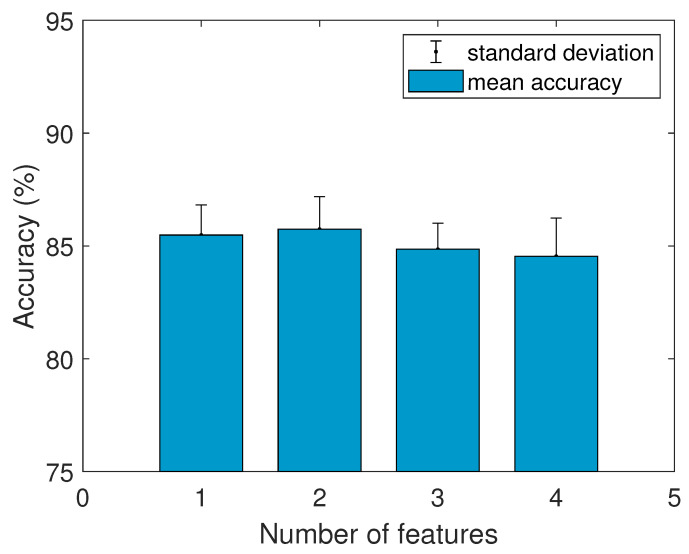
Average accuracy between real and predicted scores for MLR models using 1 to 4 features of EEG data.

**Figure 10 brainsci-11-00698-f010:**
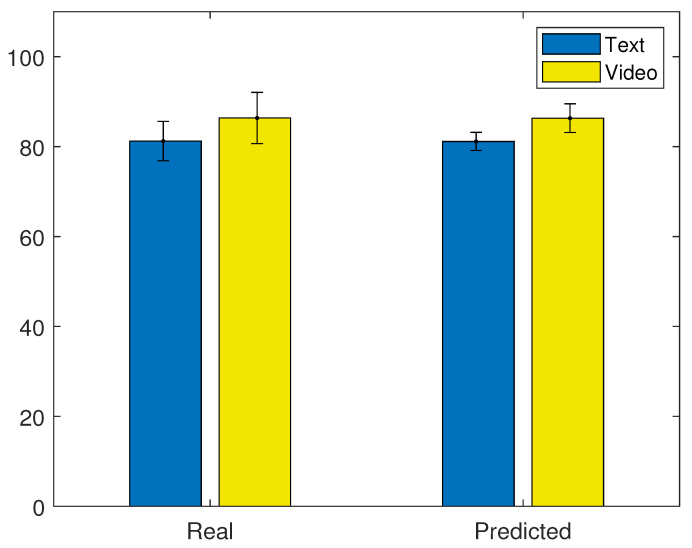
Average real and predicted scores for text and video groups across cross-validations. Black lines represent standard deviation.

**Figure 11 brainsci-11-00698-f011:**
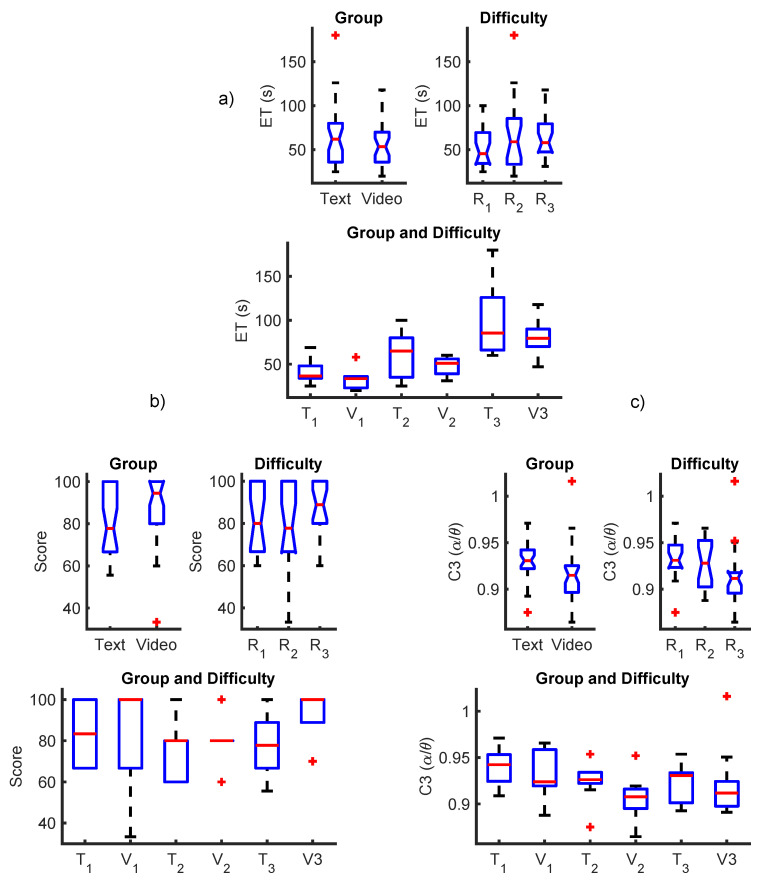
Boxplots showing the distribution of (**a**) ETs, (**b**) scores and (**c**) C3 (α/θ) across multiple factors: group (text and video), difficulty (repetitions) and both.

**Table 1 brainsci-11-00698-t001:** Frequency band and power ratio features considered for correlation analysis.

δ	θ	α	β	γ
δ/θ	θ/δ	α/δ	β/δ	γ/δ
δ/α	θ/α	α/θ	β/θ	γ/θ
δ/β	θ/β	α/β	β/α	γ/α
δ/γ	θ/γ	α/γ	β/γ	γ/β

**Table 2 brainsci-11-00698-t002:** Average scores, exam times and improvements for all learning trials.

	R${}_{1}$ Score	R${}_{1}$ ET (s)	R${}_{2}$ Score	R${}_{2}$ ET (s)	R${}_{3}$ Score	R${}_{3}$ ET (s)
Video	83.33	34.5	80	48.60	94.77	78.5
Reading	83.33	40.8	76	62.30	76.66	103.6
Improvement	0%	15%	5%	22%	19%	24%

**Table 3 brainsci-11-00698-t003:** Number of statistically significant channel (Ch) differences between text and video groups across all repetitions, for all EEG frequency bands and power ratios (features).

Feature	Ch	Feature	Ch	Feature	Ch	Feature	Ch	Feature	Ch
δ	3	θ	6	α	2	β	7	γ	3
δ/θ	0	θ/δ	0	α/δ	1	β/δ	1	γ/δ	4
δ/α	1	θ/α	2	α/θ	2	β/θ	3	γ/θ	4
δ/β	1	θ/β	3	α/β	3	β/α	3	γ/α	4
δ/γ	3	θ/γ	4	α/γ	4	β/γ	5	γ/β	5
Total	6	Total	14	Total	14	Total	19	Total	20

**Table 4 brainsci-11-00698-t004:** Selected most relevant features for the MLR model.

Feature	C3 (α/θ)	C3 (θ/α)	FP1 (δ)	C3 (γ/α)	P8 (α/γ)	P8 (γ/α)
*p*-value	0.01058	0.02139	0.03426	0.03909	0.04456	0.04994
*r* coefficient	0.33608	−0.30423	−0.28094	−0.27408	0.26712	−0.26092

**Table 5 brainsci-11-00698-t005:** Results of the Friedman tests (*p*-values) for different groups and difficulties.

Factor	Score	ET	C3 (α/θ)	C3 (θ/α)	FP1 (δ)	C3 (γ/α)	P8 (α/γ)	P8 (γ/α)
Group	0.0162	0.0490	0.0289	0.0323	0.0024	0.6622	0.4577	0.4065
Difficulty	0.2073	0.2229	0.0081	0.0079	0.0016	0.8202	0.2474	0.2284

## Data Availability

A dataset supporting this research is publicly available at https://dx.doi.org/10.21227/er9f-5b09. Accessed on 19 May 2021.

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
