# Peer review of "EEG-Based Tool for Prediction of University Students’ Cognitive Performance in the Classroom"

_brainsci, 2021, doi:10.3390/brainsci11060698_

Round 1

Reviewer 1 Report

The study evaluates the capacity of two different learning methods (video, text) to transmit knowledge to a group of students. EEG-related features were used to explore different mental processes underlying declarative learning and to predict students’ scores.

  • The introduction is complete and effective to communicate the momentum of the research to the reader. It is also well written.
  • Some sentences as the ones I cited hereafter require citing the sources they come from:

 “The theta (q: 4-7 Hz) band primarily occurs in the parietal and temporal regions of the
brain. Such waves are produced during moments of emotional pressure, interruptions of consciousness, or deep physical relaxation.”

“Workload can be inferred from the mental fatigue measurement, whereas a high cognitive workload results in a higher mental fatigue state.”

  • It is not clear how the dimensionality reduction was performed. What is the criteria used to determine the“most correlated features”? How many features remained? This preprocessing could be better explained as it is key for selecting the EEG outputs.
  • How did you verify that the subjects in the text group were reading the text? Did they read it out loud?
  • Was there any control (e.g. inclusion criteria) to verify that subjects in both groups presented similar learning capabilities? This is important because the size of the groups is reduced.
  • The reviewer wants to congratulate the authors for the addition of the supplemental It is very clear and helps understanding the manipulation.
  • Were P300 was also considered in the analysis?
  • Did you try other predictive methods besides MLR to predict student’s scores?
  • Was the proportion of users showing non-fatigued and well-rested state similar between groups? As groups sizes are small, the differences observed between groups could be strongly affected by the individual state of the users rather than the condition.

Author Response

Dear Reviewer: 

We appreciate your time in reviewing our work carefully and providing such valuable feedback. Please see the attachment; you will find a PDF with a table showing your comments, our response to them, and the location in the manuscript where they are located. Also, the editions are in bold text within the revised manuscript for a simple localization. 

Thank you

Reviewer 2 Report

In this article Authors presents a neuroengineering-based machine learning tool developed to predict students' performance under different learning modalities. Neuroengineering tools are used to predict the learning performance obtained through two different modalities: text and video. Electroencephalographic signals were recorded in the two groups during learning tasks, and performance was evaluated with tests. 

My comments to the article are as follows:

- I suggest supplementing the keywords with EEG and Cognitive performance.

- In addition, as part of the Introduction, I propose to provide a broader background in the field of EEG signal sources. In this regard, you can refer to the article: Characteristics of Question of Blind Source Separation Using Moore-Penrose Pseudoinversion for Reconstruction of EEG Signal, Automation 2017: Innovations In Automation, Robotics And Measurement Techniques, Book Series: Advances in Intelligent Systems and Computing, Springer.

It is also worth mentioning in the Introduction that there are prototypes of devices that are responsible for the verification and visualization of the concentration state on the basis of EEG. For example, you can refer to: Project and Simulation of a Portable Device for Measuring Bioelectrical Signals from the Brain for States Consciousness Verification with Visualization on LEDs, Challenges In Automation, Robotics And Measurement Techniques, Book Series: Advances in Intelligent Systems and Computing, Springer from 2016.

- I propose to separate the Results section from the Discussion. This section is now long. This will help the reader read it.

- There are no references to mathematical formulas / formulas in the text. This should be completed.

- As part of Conclusions, I propose to describe future plans for research.

Author Response

(The authors gave the same response as above.)
